# Mediation of Sinusoidal Network Oscillations in the Locus Coeruleus of Newborn Rat Slices by Pharmacologically Distinct AMPA and KA Receptors

**DOI:** 10.3390/brainsci12070945

**Published:** 2022-07-19

**Authors:** Bijal Rawal, Klaus Ballanyi

**Affiliations:** Department of Physiology, Faculty of Medicine & Dentistry, 750 MSB, University of Alberta, Edmonton, AB T6G 2H7, Canada; bijal@ualberta.ca

**Keywords:** AMPA, brain slices, iGluR, glutamate, kainate, local field potential, locus coeruleus, neuron, noradrenaline, norepinephrine, neonatal, oscillations, pattern transformation, rhythm generation, synchronization

## Abstract

Brain control by locus coeruleus (LC) neurons involves afferent glutamate (Glu) inputs. In newborns, LC Glu receptors and responses may be sparse due to immaturity of the brain circuits providing such input. However, we reported, using newborn rat brain slices, that Glu and its ionotropic receptor (iGluR) agonist NMDA transform spontaneous local field potential (LFP) rhythm. Here, we studied whether α-amino-3-hydroxy-5-methyl-4-isoxazole propionic-acid (AMPA) and kainate (KA) iGluR subtypes also transform the LFP pattern. AMPA (0.25–0.5 µM) and KA (0.5–2.5 µM) merged ~0.2 s-lasting bell-shaped LFP events occurring at ~1 Hz into ~40% shorter and ~4-fold faster spindle-shaped and more regular sinusoidal oscillations. The AMPA/KA effects were associated with a 3.1/4.3-fold accelerated phase-locked single neuron spiking due to 4.0/4.2 mV depolarization while spike jitter decreased to 64/42% of the control, respectively. Raising extracellular K^+^ from 3 to 9 mM increased the LFP rate 1.4-fold or elicited slower multipeak events. A blockade of Cl^−^-mediated inhibition with gabazine (5 μM) plus strychnine (10 μM) affected neither the control rhythm nor AMPA/KA oscillations. GYKI-53655 (25 μM) blocked AMPA (but not KA) oscillations whereas UBP-302 (25 μM) blocked KA (but not AMPA) oscillations. Our findings revealed that AMPA and KA evoke a similar novel neural network discharge pattern transformation type by acting on pharmacologically distinct AMPAR and KA receptors. This shows that already the neonatal LC can generate oscillatory network behaviors that may be important, for example, for responses to opioids.

## 1. Introduction

The locus coeruleus (LC) in the lower brainstem controls many brain functions such as sleep, memory, anxiety, breathing, pain sensation, and opioid actions or addiction/withdrawal [1,2,3,4]. Specifically, the LC acts as a hub by releasing noradrenaline into most brain areas during its (tonic) spontaneous activity and, phasically, in response to afferent (sensory) synaptic inputs from various other brain circuits [5,6,7,8,9,10,11,12,13,14]. The afferents form synapses within the LC neuron somata core and surrounding ‘pericoerulear’ areas and contain distinct neurotransmitters such as γ-aminobutyric acid (GABA), serotonin, opioids, or glutamate (Glu). Regarding Glu, axon terminals from, for example, the nucleus paragigantocellularis, posterior hypothalamus, lateral habenula, and prefrontal cortex innervate the LC [6,7,8,9,10,11]. It has been studied extensively how brain functions are modulated by the activation of the ionotropic subclass of Glu receptors (iGluR) [15] in LC neurons. For example, the N-Methyl-D-Aspartate (NMDA)-subtype iGluR antagonist ketamine has antidepressant effects [16,17,18]. Moreover, during systemic morphine application to adult rats, iGluRs on LC neurons contribute presumably to the transformation of their tonic spiking into bursting [19]. As stated in the latter report, dose-response studies are needed to find out whether both NMDA-type and non-NMDA-type iGluRs that are activated by α-amino-3-hydroxy-5-methyl-4-isoxazole propionic-acid (AMPAR) and kainate (KAR) are involved in opioid-evoked LC discharge pattern transformation. That AMPAR/KARs might be more important in that regard is indicated by the finding that intracoerulear application of their common ‘classical’ antagonist 6-cyano-7-nitroquinoxaline-2,3-dione (CNQX) is more effective than the NMDAR antagonist (2R)-amino-5-phosphonovaleric acid (AP5) in suppressing opioid-related LC neuron activation that is caused by intravenous injection of the opioid receptor antagonist naloxone [20]. The authors concluded that a substantial part of LC hyperactivity during opioid withdrawal is mediated by enhanced input from excitatory amino acids including Glu.

The above results on LC functions were obtained in adult mammals, mostly rats. In newborns, the LC is already well developed and controls other brain circuits, including support of their development [21,22,23,24,25]. Some brain structures that provide Glu input to the adult LC may still be immature at birth [24]. Thus, LC innervation with Glu afferents, and possibly also iGluR expression, may be less pronounced in newborns. There is only sparse information on the in vivo properties of the neonatal LC as this nucleus is located deep within the brainstem and contains only several thousand neurons [2,3,4,8,14,26]. As both these features and other in vivo limitations hamper pharmacological and electrophysiological analyses, particularly in newborns, brain slices are being used [22,23,24,25]. We recently showed in newborn rat slices that NMDAR activation transforms bell-shaped local field potential (LFP) events that are occurring at 1 Hz into faster burst-like multipeak signals [1]. Also AMPARs seem to be functional in the newborn rat LC as indicated by our finding [27] that CNQX may act here as a partial agonist due to their coupling to transmembrane AMPAR proteins (TARP) [28,29]. It was the aim of the present study to use the same in vitro approaches as in our latter report [1] to investigate whether AMPARs and KARs are both functional, have different pharmacological properties and may thus have distinct effects on the neonatal LC network. Such information will enhance the understanding of mechanistic neonatal LC network organization to study in the future, e.g., whether (one of) these iGluR sub-types contributes to the in vivo-like opioid-evoked slow bursting that we found in newborn rat slices [30,31]. 

## 2. Materials and Methods

### 2.1. Preparation and Solutions 

The experiments were performed on horizontal brain slices from 0–5 days-old CD-001 (SD) rats of unknown sex (Charles River Laboratory Inc., Wilmington, MA, USA). All procedures were approved by the University of Alberta Animal Care and Use Committee and in compliance with the guidelines of the Canadian Council for Animal Care and in accordance with the Society for Neuroscience’s ‘Policies on the Use of Animals and Humans in Neuroscience Research’. 

The procedures for generating LC-containing brain slices are described elsewhere in detail [32]. In brief, the rat pups were anesthetized with 2–3% isoflurane to a level that caused disappearance of the paw withdrawal reflex. They were then decerebrated and the neuraxis was isolated at 18–20 °C in superfusate containing (in mM): 120 NaCl, 3 KCl, 1.2 CaCl_2_, 2 MgSO_4_, 26 NaHCO_3_, 1.25 NaH_2_PO_4_, and 10 D-glucose (pH adjusted to 7.4 by gassing with carbogen, i.e., a mixture of 95% O_2_ plus 5% CO_2_). 

The brain was glued on its ventral surface to a metal cutting plate which was then inserted into the vise of a vibratome (Leica VT1000S; Leica Microsystems, Richmond Hill, ON, Canada). In carbogenated superfusate, horizontal brain slices were cut at room temperature, initially at 400–600 µm steps, until the 4th ventricle appeared. Once the LC appeared as a dark oval area close to the lateral border of the 4th ventricle, a single 400 µm thick slice was cut. This slice typically contained >50% of the dorsoventral aspect of the approximately circular LC somata area which extends in ratsby ~300 µm in the horizontal plane [33]. At this section level, the neonatal rat LC is not surrounded by, or intermingled with, another obvious brain nucleus. It can thus be identified as a dense cluster of large neuron somata with a diameter of ≥20 µm as illustrated in our previous reports [27,30,31,32]. 

For electrophysiological recording, a slice was mechanically fixed with a platinum ‘harp’ in an acrylic chamber (volume ~1 mL) with a glass bottom (Warner Instruments, Hamden, CT, USA). The LC neurons were visualized with a 20 × objective (XLUMPlanF1, numerical aperture 1.0) that was attached to an MPE microscope (Olympus, Markham, ON, Canada) or an IR-DIC video camera (OLY-150, Olympus). A peristaltic pump (Sci-Q 403U/VM, Watson-Marlow, Wilmington, MA, USA) was used to apply carbogenated superfusate at a rate of 5 mL/min which was removed from the distal aspect of the chamber with a vacuum that was applied to a hypodermic needle. The superfusate temperature in the chamber was kept at 28 °C via a heat control system (Thermo-Haake DC10-V15/B, Sigma-Aldrich, Markham, ON, Canada). 

### 2.2. Agents and Drug Application

The agents (stock solutions) were: AMPA (1 M in H_2_O); CNQX (100 mM in DMSO); GYKI-53655 (25 mM in H_2_O); KA (1 mM in H_2_O); kynurenic acid (100 mM in DMSO); (1,2,3,4-Tetrahydro-6-nitro-2,3-dioxo-benzo[f]quinoxaline-7-sulfon-amidehydrate (NBQX, 100 mM in DMSO); UBP-302 (25 mM in H_2_O); gabazine (25 mM in H_2_O); and strychnine (10 mM in H_2_O). All the drugs were bath-applied (AMPA and KA for 5 min each). AMPA and KA were bath-applied for a comparison with previous findings on LC properties in rodent brain slices. Moreover, bath-application of KA is a standard approach to elicit stable gammaoscillations in brain slices containing other neural circuits, with the hippocampus most frequently studied [34,35,36,37,38]. This enabled a comparison of the present findings with those from these studies. Note that the wash times varied depending on how quickly the effect of a drug was reversible.

### 2.3. Electrophysiological Recording

Patch pipettes were pulled from borosilicate glass capillaries (GC-150TF-10; 1.5 mm outer Ø, 1.17 mm inner Ø, Harvard Apparatus) to an outer tip Ø of ~2 μm using a vertical puller (PC-10, Narishige International Inc., Amityville, NY, USA). They were used to record the LFP (after breaking and beveling the tip to obtain a suction electrode) or membrane potential (V_m_) in a single LC neuron. Electrophysiological signals were sampled at 4–20 kHz into a digital recorder (Powerlab 8/35 + LabChart 7 software, ADInstruments, Colorado Springs, CO, USA) that was connected to a personal computer. 

Suction electrode recording: The major aim of this study was to analyze AMPA- or KA-evoked transformation of the LFP pattern. We reported [30,32] that a robust LFP can be recorded with suction electrodes that are often used to monitor nerve activities or neuronal population bursting within the isolated breathing center of newborn rodent brainstem slices [39]). For such recording, patch pipettes were broken and subsequently beveled manually with sandpaper (Ultra Fine 600 Grit, Norton-Saint Gobain, Worcester, WA, USA) at an angle of 45° to an oval-shaped tip with an inner opening of 40–60 μm. After filling with superfusate, the dc resistance of the suction electrodes was ~200 kΩ [30,32]. Following insertion into a patch electrode holder system (ESP-M15P and MHH-25, Warner Instruments), 15–30 mmHg suction was applied to the electrode with a syringe (BD Diagnostics, Franklin Lakes, NJ, USA) and controlled via a differential pressure sensor (Honeywell, SCX05DN, Fort Worth, TX, USA). An MP-285 micromanipulator (Sutter Instrument Company, Novato, CA, USA) was used to position the electrode opening at an angle of ~30° on the slice surface followed by the application of <5 mmHg suction. The ‘raw’ suction electrode signal was amplified (x10 k) and band-pass-filtered (0.3–3 kHz) using a Model-1700 differential amplifier (AM-Systems, Sequim, WA, USA). In parallel, the signal was processed for another recording channel by integration (‘moving average’) with a time constant of 50 ms using a MA-821/RSP unit (CWE, www.cwe-inc.com, accessed on 25 May 2022). The integrated signal recording is used, for example, for respiratory [39] or locomotor [40,41] network analyses.

Whole-cell recording: For neuronal V_m_ recording using an EPC-10 amplifier (HEKA Lambrecht, Germany), patch pipettes were filled with (in mM): 140 K-gluconate, 1 NaCl, 0.5 CaCl_2_, 1 MgCl_2_, 1 Na_2_-ATP, 1 mM BAPTA, and 10 Hepes (pH was adjusted to 7.4 with KOH; dc resistance in superfusate was 5–8 MΩ). When dimpling of the soma area occurred while visually targeting a neuron using the same manipulator as for the extracellular recording, 20 mmHg positive pressure was released and negative pressure was applied for gigaseal formation (>1 GΩ). 

Whole-cell V_m_ recording was established by abrupt suction (~100 mmHg). Series resistance, comprising access, plus electrode resistance was compensated during a test pulse at the beginning of a recording and was also checked, and eventually adjusted later during the measurement. The access resistance typically ranged between 10–50 MΩ and was stable in >90% of neurons even during recordings lasting >1 h. For determining the neuronal input resistance ranging from 120–370 MΩ, hyperpolarizing current pulses (50–100 pA) were injected, mostly at an interval of 10–15 s. Only cells were analyzed in which the spike amplitude was >70 mV while the V_m_ was stable for a 5 min control period. Due to ongoing subthreshold oscillations (STOs) [22,31], LC neurons do not have a ‘resting’ V_m_ which was thus defined as the value at 50% of the interval between spikes at the oscillation peak. 

### 2.4. Data Analysis

LFP rates and amplitudes were quantified during 1 min recording time periods in the control or at steady-state of drug effects. LFP event duration was defined with Clampfit software (Molecular Devices Corporation, Chicago, IL, USA) as the time interval from when the averaged signal increased above and decreased below a threshold that was set at 10% of the peak amplitude, respectively. The extent of LC network synchrony was determined with Clampfit software by comparing over a time period of 10 s the cross-correlation between the whole-cell-recorded single neuron spiking and the peak of the integrated LFP signal. The peaks in the cross-correlograms refer to the cross-correlation function estimate (CFE) values for the accuracy of synchrony. The lag time quantifies the shift in the peak between these events and thus gives a measure of spike jitter. This approach is applied to correlate single neuron with (nerve) population bursting in respiratory [42] and locomotor networks [40]. For analysis of spike jitter, 20 cycles of LFP events and neuronal spiking were temporally aligned to the LFP peak using Clampfit software. The aligned traces were then overlapped and analyzed using a numerical matrix technique (Origin 6, Microcal Software). A summated LFP was obtained by averaging the 20 cycles. The average time point of neuronal spiking and its SD was used as a measure of jitter of their discharge.

The regularity of the LC network rhythm was determined by quantifying the irregularity (IR) score which is an established parameter to analyze rhythmic neural network bursting, e.g., of the inspiratory center [43,44] using the formula: IR score = 100 * [Σ(P_*n*_ − P_*n*−1_)/P_*n*−1_]/N, where N is the number of events, P_*n*_ is the period of *n*th event, and P_*n*−1_ is the period of the preceding discharge. Note that the IR score has no unit. The lower the value, the more regular the network rhythm is.

Values are given as means ± SD and *n*-values correspond to measurements in 1 slice per animal. Significance (non-significant [ns]: *p* > 0.05, * *p* < 0.05, ** *p* < 0.01, *** *p* < 0.001, **** *p* < 0.0001) was assessed by two-tailed paired t-test and a one-way ANOVA with Dunnett’s post-test (only applied for analysis of AMPA and KA effects on LFP) using Prism software (GraphPad Software Inc., La Jolla, CA, USA). To facilitate reading of the Results section text, quantified data that are shown with statistical details in the scatter plots of the Figures are only represented as means ± SD.

## 3. Results

Initial analysis of the LFP properties and dose-dependent LFP pattern changes upon 5 min AMPA or KA application was followed by cellular analyses with combined LFP and V_m_ recording.

### 3.1. AMPA-Evoked LFP Oscillations

We reported that LFP events in the LC of newborn rat slices are typically bell-shaped due to a normal distribution of single spikes whose sum jitters equally around the LFP peak [1]. Here, firstly dose-response relationships were determined for AMPA and KA effects. This served to identify thresholds for a putative LFP pattern transformation and to elucidate whether high agonist concentrations perturb LC network rhythm. 

AMPA effects on a single slice are exemplified in Figure 1 while statistical analysis is shown in Figure 2. AMPA had no effect at 0.05 µM whereas 0.1 µM accelerated the LFP rate ~1.5-fold (Figure 1A). The LFP rate increase was stronger at 0.25 µM AMPA and single events merged into sinusoidally-shaped ones that showed every 2–3 s amplitude fluctuations by 10–40% resulting in a spindle-shaped LFP envelope that was particularly evident in the integrated signal trace (Figure 1B). Similar effects were seen at 0.5 µM, but the LFP amplitude declined progressively during AMPA, even into the early phase of wash (Figure 1C). The amplitude fluctuations during AMPA-evoked LFP oscillations were due to the occurrence of tonic spike discharge that was evident in the raw trace and could raise the baseline of the integrated trace if prominent (Figure 1C,D). At 1 µM, AMPA evoked initially similar spindle-shaped LFP oscillations, but the rhythm was blocked before the end of application (Figure 1E). 

As analyzed for five slices in Figure 2, AMPA accelerated the LFP rate from 53.6 ± 20.9 events/min (i.e., 0.89 ± 0.34 Hz) to 225 ± 54.2 events/min (i.e., 4.5 ± 1.6-fold) at 0.25 µM and to 305 ± 80.7 events/min (i.e., 5.1 ± 0.8-fold) at 0.5 µM (Figure 2A). Correspondingly, single LFP events were shortened from 157 ± 16.0 ms to 106 ± 22.8 ms (i.e., 68.2 ± 17.6% of the control) at 0.25 µM and to 81.8 ± 18.0 ms (i.e., 54.6 ± 16.6% of the control) at 0.5 µM (Figure 2A). The single LFP event amplitude decreased to 73.0 ± 4.0% of the control at 0.25 µM and to 58.0 ± 13.0% of the control at 0.5 µM (Figure 2A). The bell-shape of the LFP event was maintained during AMPA (Figure 2B). The effects of 1 µM AMPA were not quantified as the rhythm was blocked too quickly. Within 11.0 ± 4.0 min after the start of washout, the LFP recovered after depression or blockade. At intermediate AMPA doses that do not block the rhythm, recovery occurred with no arrest of discharge early during washout as seen in response to the bath-applied glutamate or N-methyl-D-aspartate in our recent study [1]. The original recordings already indicated that 0.25 and 0.5 µM AMPA made the LFP rate more regular. Indeed, as quantified in six slices for 0.25 µM AMPA, the irregularity score of rhythm decreased from 60.1 ± 9.7 in the control to 9.2 ± 5.2 (Figure 2C).

### 3.2. KA-Evoked LFP Oscillations

KA effects on a single slice are exemplified in Figure 3 while statistical analysis in five slices is shown in Figure 4. At 0.05 and 0.1 µM, KA had no effect while 0.25 µM accelerated the rate of still separate LFP events that merged at 1 µM into sinusoidally-shaped oscillations with 10–40% amplitude fluctuations such as those that were seen during AMPA (Figure 3A–C). At 2.5 µM, KA evoked similar spindle-shaped oscillations and the integrated signal trace baseline increased slightly due to tonic spiking (Figure 3C,D). At 5 µM, KA initially caused spindle-shaped LFP oscillations and an integrated signal baseline rise before the rhythm was depressed and then blocked after 4.5 min (Figure 3D).

As summarized in Figure 4, KA increased the LFP rate from 50.7 ± 12.7 events/min (i.e., 0.84 ± 0.21 Hz) to 134 ± 32.4 events/min (i.e., 2.7 ± 0.6-fold) at 0.5 µM, to 181 ± 36.3 events/min (i.e., 3.6 ± 0.6-fold) at 1 µM and to 282 ± 88.8 events/min (i.e., 5.8 ± 2.5-fold) at 2.5 µM (Figure 4A). Correspondingly, single LFP event duration decreased from 186 ± 48.8 ms to 133 ± 18.9 ms (i.e., 74.2 ± 16.3% of the control) at 0.5 µM, to 127 ± 15.4 ms (i.e., 70.9 ± 13.5% of the control) at 1.0 µM and 107 ± 24.8 ms (i.e., 60.2 ± 17.6% of the control) at 2.5 µM (Figure 4A). The LFP event amplitude decreased at 1.0 µM to 77.9 ± 11.8% of the control and at 2.5 µM to 66.3 ± 14.2% of the control (Figure 4A). The bell-shape of the integrated LFP signal remained even during 2.5 µM KA (Figure 4B). The effects of 5 µM KA were not quantified as rhythm was quickly blocked in all five slices, but recovered within 4.5 ± 0.9 min after the start of washout. At intermediate KA doses that were not blocking the rhythm during application, no depression occurred early during washout, similar to AMPA. As was quantified for 2.5 µM, KA decreased the irregularity score from 56.4 ± 25.6 in the control to 7.8 ± 3.6 (*n* = 5) (Figure 4C).

### 3.3. Combined LFP and V_m_ Analysis of AMPAR Agonist Effects 

The finding that both AMPA and KA shortened LFP event duration suggests that the agents reduce spike jitter and increase LC network discharge synchrony. This was studied with a combined LFP and V_m_ recording. During LFP acceleration and pattern transformation in 0.25 µM AMPA, seven LC neurons depolarized from −44.0 ± 2.3 mV by 4.0 ± 1.0 mV (*p* < 0.01) and increased their regular firing rate 3.1-fold from 56.1 ± 12.5 spikes/min to 175.1 ± 68.0 spikes/min (Figure 5 and Figure 6A). Spike jitter (Figure 6B) that was analyzed in five of the neurons, decreased from ±89.3 ± 24.2 ms to ±61.4 ± 26.3 ms (i.e., to 66.5 ± 12.0% of control) similar to shortening of the LFP event in these slices (Figure 6C). 

During LFP acceleration and pattern transformation by 2.5 µM KA, seven neurons depolarized from −47.1 ± 2.1 mV by 4.5 ± 1.5 mV (*p* < 0.01) and increased their regular firing rate 4.3-fold, from 63.6 ± 22.6 spikes/min to 272 ± 78.8 spikes/min (Figure 6D and Figure 7). Spike jitter (Figure 6E) that was determined in five of the neurons, fell from ±71.5 ± 34.5 ms to ±24.2 ± 11.1 ms (i.e., to 37.9 ± 19.1% of control) similar to the shortening of the LFP event in these slices (Figure 6F). 

Cross-correlation analysis was attempted by comparing the peak of the integrated LFP signal with intracellularly recorded spiking. In the seven neurons that were tested for AMPA, the mean control values for CFE and lag time were, respectively, 0.31 ± 0.08 and 112.8 ± 28.1 ms. The corresponding control values for the seven neurons that were tested for KA were, respectively, 0.36 ± 0.05 and 109.3 ± 21.6 ms. The numbers could not be determined in any of these neurons during AMPA or KA as the cross-correlograms did not show quantifiable values. 

As exemplified in Figure 5 for one of four neurons, the non-selective iGluR blocker kynurenic acid (0.5 mM) [15] restored V_m_ during 0.25 µM AMPA from −41.2 ± 1.1 mV to −45.2 ± 1.2 mV (*p* < 0.05) and the firing rate from 192.5 ± 82.2 to 50.2 ± 19.5 spikes/min (*p* < 0.05). In four different neurons, kynurenic acid restored V_m_ during 2.5 µM KA from −42.2 ± 2.5 mV to −46.7 ± 2.6 mV and AP rate from 290 ± 87 spikes/min to 85.0 ± 28.0 (not shown). In all eight neurons, the agent also reversed the spindle-shaped oscillatory LFP pattern to the control (Figure 5).

### 3.4. Effects of Increased Network Excitability via Raised Extracellular K^+^

Enhanced (rhythmic) neuronal activity raises extracellular K^+^ and vice versa [39,45,46,47]. Here, the effects of raising superfusate K^+^ from physiological 3 mM to 5, 7, and 9 mM were tested on the LFP pattern (Figure 8A). Rhythm in 5 mM (*n* = 6) and 7 mM (*n* = 9) K^+^ did not change, except in one slice for 7 mm K^+^ where the LFP pattern transformed to slower multi-peak events with a mean increase in event duration from 232 ± 31.4 ms to 965 ± 365 ms. In 11 slices, the effects of 9 mM K^+^ were tested. In seven cases, the LFP rate increased to 143 ± 33.6% of the control (i.e., from 60.0 ± 20.7 events/min to 91.4 ± 10.9 events/min or 1.00 ± 0.34 Hz to 1.52 ± 0.18 Hz) with no effect on the burst duration or amplitude (Figure 8B). In the other four cases, the LFP pattern transformed into slower and longer multi-peak events (Figure 8C). Specifically, 9 mM K^+^ slowed the LFP rate to 30.1 ± 8.7% of the control (i.e., from 60.0 ± 20.7 events/min to 16.6 ± 7.27 events/min or 1.0 ± 0.34 to 0.28 ± 0.12 Hz) (Figure 8D). The LFP event duration increased to 648 ± 107% of control (i.e., from 372 ± 119 ms to 2514 ± 755 ms) with no effect on the burst amplitude (Figure 8D). All the parameters recovered within <5 min after the start of the washout of raised K^+^.

### 3.5. Effects of Specific Blockers of Anion Channel-Mediated Inhibition

In hippocampus or cortex slices, the induction of (gammatype) spindle-shaped LFP oscillations by bath-applied KA involves interneurons that exert a tonic inhibitory influence via Cl^−^-mediated inhibition due to GABA_A_ and/or glycine receptors [35,36,37]. Thus, it was studied next if AMPA/KA oscillations or control rhythm are affected by the blockade of these receptors with gabazine (5 μM) plus strychnine (10 μM), respectively.

In nine slices, the blockers had no effect on the LC rhythm (Figure 9A–C). Specifically, the LFP rate in the control vs. gabazine plus strychnine was 0.98 ± 0.20 Hz vs. 0.98 ± 0.18 Hz (or 58.7 ± 12.0 events/min vs. 58.9 ± 10.8 events/min (102 ± 17.5% of control). The LFP event duration was 317 ± 67.5 vs. 327 ± 74.4 ms (104.1 ± 15.9% of the control. The LFP amplitude was 101 ± 11.2% of the control.

The addition of AMPA (*n* = 7) (Figure 9A,C,D) or KA (*n* = 6) (Figure 9B–D) to gabazine- plus strychnine-containing superfusate evoked the typical LFP oscillations at their normal doses, i.e., 0.25 or 0.5 µM, and 0.5 or 1 µM, respectively. For AMPA, the LFP rate increased from 0.93 ± 0.14 to 3.80 ± 0.97 Hz (or 55.7 ± 8.67 events/min to 228 ± 58.0 events/min, i.e., to 414 ± 111% of control). The LFP event duration decreased from 303 ± 52.2 ms to 151 ± 39.0 ms (i.e., to 50.0 ± 9.6% of control) whereas its amplitude decreased to 43.3 ± 18.7% of the control. For KA, the LFP rate increased from 0.89 ± 0.30 Hz to 2.90 ± 0.41 Hz (or from 53.5 ± 18.1 to 174 ± 24.4 events/min, i.e., 343 ± 65.7% of the control). The LFP event duration decreased from 291 ± 81.3 ms to 164 ± 20.3 ms (i.e., 59.8 ± 17.2% of control) whereas its amplitude decreased to 66.0 ± 12.1% of the control). This showed that a comparable concentration of both agents had the same significant effects on these parameters in the presence of the GABA_A_ and glycine receptors blockers (shown here) as in the experiments that were done in their absence (Figure 2A and Figure 4A).

### 3.6. Effects of iGluR Blockers on LFP Oscillations

In a further experimental approach, it was investigated whether the very similar AMPA- and KA-evoked spindle-shaped LFP oscillations show a different sensitivity to the specific competitive AMPAR blocker GYKI, the specific KA receptor (KAR) blocker UBP-302, and the non-selective AMPAR/KAR blockers CNQX and NBQX [15]. At an AMPA or KA dose that evoked persistent LFP oscillations either in standard superfusate or in the presence of gabazine plus strychnine, the blockers were added. The data were pooled as gabazine plus strychnine affected neither the control rhythm nor AMPA/KA-evoked LFP oscillations as shown in the preceding paragraph. Only the effects on the LFP event rate were analyzed as the most prominent change that was evoked by the agonists.

As shown in Figure 10A,B, AMPA increased the LFP rate from 0.93 ± 0.15 to 3.99 ± 0.88 Hz (or from 55.7 ± 9.16 events/min to 239 ± 52.9 events/min, i.e., to 435 ± 104% of the control, *n* = 6). The addition of 5 µM GYKI left the LFP ocillations unchanged at 4.08 ± 0.51 Hz (or 245 ± 30.9 Hz, i.e., 450 ± 114% of control) whereas 10 µM countered them with LFP rate decreasing to 1.22 ± 0.75 Hz (or to 73.0 ± 44.7 events/min, i.e., 120 ± 71.4% of the control) and 25 µM to 0.75 ± 0.17 Hz (or to 45.2 ± 10.4 events/min, i.e., 82.5 ± 14.3% of the control). The GYKI effects were irreversible at least for 30 min as the rhythm remained at control rhythm values of 0.80 ± 0.17 Hz (or 48.1 ± 10.2 events/min; i.e., 88.2 ± 17.8% of control) upon wash of GYKI in AMPA. 

As shown in Figure 10C,D, UBP failed to counter the AMPA-evoked fast LFP oscillations in four different slices. Specifically, AMPA increased the LFP rate to 3.94 ± 1.46 Hz (or 236 ± 87.7 events/min, i.e., to 280 ± 110% of the control). During 25 µM or 100 µM UBP, the LFP rate remained elevated, namely at 307 ± 144% of the control and 193 ± 60.6% of the control. A wash of UBP did also not reduce the LFP rate (198 ± 46.6% of the control) whereas subsequent wash of AMPA restored LFP rate to a value of 62.6 ± 20.3% of the control. 

We previously reported that CNQX has a partial agonistic action on LFP rhythm [27]. Here, the addition of 100 µM CNQX to the control superfusate increased the LFP rate in five slices to 245 ± 69.1% of the control (or from 64.8 ± 19.0 events/min to 152.4 ± 35.7 events/min, i.e., 1.08 ± 0.32 to 2.54 ± 0.59 Hz). Only in one of these cases did the LFP pattern transform into spindle-shaped bursts. In the presence of 100 µM CNQX, neither 0.25 µM nor 0.5 µM AMPA (*n* = 5 each) or even 1 or 2.5 µM AMPA (*n* = 4 each) (Figure 10E) were able to evoke spindle-shaped LFP oscillations (*p* = ns for rate, amplitude, and duration) (data not shown). In two other slices, 100 µM CNQX reversed the AMPA-evoked spindle-shaped LFP oscillations (Figure 10F).

As shown in Figure 11A,B, KA increased the LFP rate in five different slices from 1.38 ± 0.52 to 2.75 ± 0.42 Hz (or 82.7 ± 31.2 events/min to 165 ± 25.5 events/min, i.e., to 222 ± 86.2% of the control). UBP at 5 or 10 µM did not affect the LFP rate remaining at 209 ± 82.0% of the control and 156 ± 78.3% of the control, respectively. At 25 µM, UBP restored the LFP rate to 91.1 ± 25.9% of the control. As shown in Figure 11C,D, in eight slices, KA-evoked LFP oscillations (2.74 ± 0.46 Hz or 164 ± 27.5 events/min vs. 0.90 ± 0.23 Hz or 54.0 ± 13.5 events/min in the control, i.e., 333 ± 65.6% of the control) were affected neither by 25 µM GYKI at a rate of 319 ± 103% of the control nor by the further addition of 25 µM CNQX at a rate of 265 ± 88.8% of the control. At 50 µM, CNQX in GYKI reversed the KA-evoked increase in the LFP rate to 182 ± 55.4% of the control whereas 100 µM of the agent decreased the LFP rate to 121 ± 25.4% of the control (Figure 11E). Wash of CNQX in GYKI restored the KA-evoked LFP oscillation to 299 ± 40.2% of the control within <10 min (Figure 11D). 

In a final set of experiments, the addition of NBQX to superfusate had no effect on the control LFP at doses of 5, 10, or 100 µM (*n* = 5 each). However, 2.5 µM (*n* = 3) or 5 µM (*n* = 2) NBQX reversed the AMPA-evoked LFP oscillations whereas 10 µM (*n* = 6), 25 µM (*n* = 3) or even 100 µM (*n* = 3) were necessary to counter the KA-evoked LFP oscillations (data not shown).

## 4. Discussion

We found that AMPAR and KAR with distinct pharmacological properties are functional in the neonatal rat LC and can transform separate spontaneous LFP events in slices into faster, more regular, and merged spindle-shaped oscillatory events. The mechanisms and consequences for LC network modeling and functions are discussed.

### 4.1. Complex-Mediated Spindle-Shaped Faster and More Regular Network Rhythm 

We previously reported [27] that the frequently used AMPAR blocker CNQX makes the neonatal rat LC population rhythm faster and more regular by acting as a partial agonist on a TARP-AMPAR complex that is also functional in other neurons [28,29]. In some slices of our above report, CNQX merged separate LFP bursts into spindle-shaped sinusoidal oscillations. Here, we found that such oscillations occur in all slices upon bath-application of the commonly used AMPAR agonists AMPA and KA. 

Both AMPA and KA appeared to increase the LC network synchrony as indicated by shortening of the duration of single LFP events and a corresponding reduction of single neuron spike jitter. Cross-correlation analysis between LFP and spiking in seven neurons that were tested each for AMPA or KA revealed a small degree of synchrony in the control solution based on low mean cross-correlation function estimate values. Similarly, low control values were found in our related studies in which the LFP duration was shortened by CNQX [27] or NMDA [1]. Indeed, NMDA increased LC network synchrony in our latter study, whereas no cross-correlation function estimate value could be quantified previously for any neuron during CNQX [27], as also found here for both AMPA and KA. Thus, a different type of (correlation) analysis needs to be applied in future studies testing AMPAR or KAR activation; e.g., activities of neuron pairs can be correlated as in other network synchrony studies [25,48,49,50,51,52]. Nevertheless, it can be concluded that TARP-AMPAR complex activation increases the phase-lock of LC spiking as both agonists decreased the LFP burst duration and single neuron spike jitter. 

Stable LFP acceleration and oscillatory pattern transformation occurred mostly in 0.25 µM AMPA and 2.5 µM KA, which increased its rate 4.5- and 5.8-fold, respectively. These LFP rate increases were reflected by 3.1- and 4.3-fold faster neuronal spiking due to depolarization by 4.0 and 4.2 mV, respectively. A similar dose-dependence of AMPA- and KA-enhanced single LC neuron spiking was found with extracellular recording in adult rat brain slices [53,54]. Increasing the AMPA and KA doses to 0.5 µM and 5 µM, respectively, initially had effects on the LFP that were similar to those of 0.25 µM vs. 2.5 µM. However, within several minutes following start of the 5 min application period, tonic discharge developed and depressed the rhythm. At 1 µM AMPA and 5 µM KA, the tonic discharge plus depression of the rhythm occurred earlier and the rhythm was subsequently blocked. The appearance of tonic discharge indicates that some neurons became uncoupled from the network to cause a further decrease in the extent of spike synchronization. As noted above, this needs to be analyzed with a different approach than what was used here. The subsequent blockade of the rhythm indicates a major neuronal depolarization leading to the inactivation of spontaneous Na^+^ spikes. 

In adult rat slices, extracellular recording showed that accelerated spiking of single LC neurons persisted during prolonged exposure to glutamate but was followed by post-agonist depression (PAD) of discharge early during wash [54,55]. A similar PAD was seen after the enhancement of spiking in LC neurons in vivo [56] and in our related study early during recovery from glutamate and NMDA [1]. Here, the rhythm recovered to the control within several minutes after the start of washout without such depression in response to intermediate AMPA and KA doses, causing a stable LFP pattern transformation. 

As one possible explanation for the AMPA- and KA-evoked sinusoidally-shaped oscillations, this transformed LFP pattern might represent a summation of enhanced STOs in LC neurons that are strongly electrotonically coupled in newborn rats [22,23,25]. Indeed, neurons of the inferior olive, which shares various properties with the neonatal LC such as strong coupling via gap junction, show rhythmic variations in the frequency and amplitude of single neuron STOs [48]. As these STO variations were blocked by CNQX in the latter juvenile rat brain slice report, as observed also in the present study, the authors concluded that tonic glutamatergic input is involved in this phenomenon. However, here V_m_ recording revealed that individual neurons show only an increase in the regular rate of their spiking with no rhythmic (spindle-shaped) amplitude changes for either spikes or the underlying STOs. Moreover, AMPA and KA effects were reversed here by the non-selective competitive iGluR antagonist kynurenic acid as an indication of a mechanism that differs from that in the inferior olive. Similarly, the non-competitive AMPAR antagonist GYKI countered the stimulatory CNQX effects in a recent report [27], and in the present study also blocked the AMPA-induced, but not KA-evoked LFP oscillations. Conversely, in the present study the KAR blocker UBP countered the KA-evoked, but not the AMPA-induced LFP oscillations. Moreover, CNQX acts not only as a partial agonist on the neonatal rat LC, as noted above, but it is also capable of blocking the AMPA- and KA-evoked oscillations. Also, NBQX was able to block the LFP oscillations, but did not act as a partial agonist. In summary, the pharmacology in the present study describes, for the first time, the distinct effects of all these non-NMDA-type iGluR antagonists on a spontaneously active neural network.

Our results show that AMPAR agonists can evoke a major transformation of the LFP pattern in the neonatal LC as an indication of its intrinsic capability to generate different types of discharge patterns. In addition, as the control LFP rhythm persisted in kynurenic acid as in our related reports [1,27], it is evident that the mechanism of generating the normal pattern of bell-shaped burst does not depend on iGluR. This contrasts with the established findings that many neural network rhythms such as those in the inspiratory center, spinal locomotor networks, inferior olive, or cortical structures are abolished by iGluR antagonists [31,39]. Here, we also found that the LFP rhythm persists during blockade of anion channel-mediated inhibition that is needed for normal rhythmic activity in various neural networks. For example, in the locomotor system and cortical or hippocampal networks, blockade of GABA_A_ and/or glycine receptors evokes seizure-like activities [34,36,37,38,39,57,58]. The fact that the LFP pattern transformation during AMPA or KA persists during inhibition of these receptors indicates that, in addition to the notably slower rate, this type of discharge is not similar to gamma oscillations that occur in vivo as an indication of higher brain functions, e.g., in the cortex or hippocampus [34,35,36,37,38,59,60]. As shown in several of the latter studies, a further difference is that in the latter structures, application to brain slices of KA, but not AMPA mimics the (spindle-shaped) in vivo gamma oscillations in the cortex and hippocampus. A different mechanism is also suggested by the fact that gamma oscillations can be evoked in cortical and hippocampal brain slices by raised extracellular K^+^ [45,46,47], whereas the LFP pattern either slightly accelerates or transforms in the neonatal rat LC slices into slower and prolonged multipeak bursts. 

### 4.2. Consequences for Modeling of Neonatal LC Network Properties and Function

In the mature brain, the LC controls a variety of behaviors due to its wide-spread innervation of the neuraxis [2,3,4,12,13,26,61,62]. Besides, the LC plays a pivotal role in brain development as has been reviewed thoroughly [23,24]. In summary of the latter studies, rat LC neurons undergo their final division and become functional 9–12 days before birth and innervate multiple yet immature brain regions to support their functional differentiation. For this purpose, they rhythmically release NA in a highly coordinated fashion that resembles the pulsatile blood supply by the heart (for references, see [12,13]. In the newborn rat LC, pulsed NA release occurs presumably synchronously in the different target areas. As elaborated in a series of brain slice studies by Williams and colleagues, gap junction-mediated electrical coupling, particularly of the peri-coerulear dendritic region in this nucleus, is a major factor in the proposed underlying synchronous electrical activity of neonatal LC neurons [22,24,25]. However, as mentioned in two of these reports [22,24], STOs are highly synchronous in the neonatal LC whereas the discharge of a single spike at their peak is often not. Accordingly, it was previously reported by us based on extracellular recordings that the ~0.2 s-lasting LFP burst duration already shows that spiking in the newborn rat LC is not synchronous, but rather occurs in single neurons in a particular phase of the LFP and shows a jitter [30]. Here, we found a corresponding jitter that was based on V_m_ recording and also unraveled that this jitter is reduced during both AMPA and KA. As stated above, each neuron increases its spike rate during such TARP-AMPAR complex activation in regular fashion without occurrence of spindle-like STO amplitude or frequency variations. Accordingly, the target areas in the newborn rat brain are likely activated by individual LC neurons in similar fashion as in the control, only at a higher rate which enhances NA release. It needs to be analyzed in future studies by modeling plus further pharmacological manipulation which specific modulatory or feedback effects the spindle-shaped LFP oscillations within the LC might have on this network Regarding modeling, approaches such as in the inferior olive may be applied. Specifically, it has been shown that two principal characteristics of its neurons, i.e., STOs and electrical gap junctions, make this system a powerful encoder and generator of spatio-temporal patterns with different but coordinated oscillatory rhythms [63]. 

### 4.3. Potential Role of AMPAR and KAR in Neonatal LC

Our findings do not provide details about which of the afferent brain structures might already be mature enough at birth to provide Glu input to the (pericoerulaer areas of the) LC (for references, see Introduction). This is related to the fact that we used here, for a direct comparison of the results, the same techniques as in our previous report studying glutamate and NMDA roles [1]. This included bath-application of AMPA and KA which revealed that the entire LC neuron network can transform its activity in response to the uniform activation of these receptors. Under physiological conditions in vivo, the adult LC can generate a variety of activity patterns as reviewed recently [12,13]. The latter studies also noted that it has been long proposed that the adult LC releases NA in a pulsatile fashion to the numerous remote target brain areas and that this activity can be modulated by phasic sensory inputs. The authors also stated that this view was too simple as the adult LC can generate a diversity of (spontaneous) activity patterns including a slow rhythm that resembles the LFP rhythm in the newborn rat slices. So far, it has not been studied which activity patterns the neonatal LC can generate in vivo under physiological conditions, how the tone of LC neuron spiking is modulated by (sensory) inputs from remote afferents or (neighboring) interneurons and intrinsic neuromodulators. Regarding intrinsic neuromodulators, we noted recently [31] that the LFP rhythm in newborn rat slices is not depressed by blockers of μ-opioid and (auto) α_1_, α_2_, or β NA receptors. This indicates that, in addition to Glu acting on iGluR, GABA acting GABA_A_R, or the activation of glycine receptors, neither NA or endogenous opioids are involved in generating the pattern of this spontaneous activity. Here, we can only hypothesize that, in the intact LC, specific Glu input to neurons of one of the presumptive modules (for references, see [31] would evoke the oscillations mainly in this subnetwork.

As one functional role of AMPAR and KAR in the neonatal LC, they might contribute to responses to opioids. As discussed by us [31], similar slow and prolonged neuronal bursts are seen during opioid administration to the LC of newborn rat slices [30] and adult rats in vivo [19,64]. Also, morphine-evoked enhanced LC neuron (burst) activation is blocked by the inhibition of iGluR with non-NMDAR antagonists being more effective than NMDAR antagonists [20]. Accordingly, it would be interesting as a further project in slices to study whether the LFP pattern transformation during opioid application would be abolished by the selective AMPAR blocker GYKI or the specific KAR blocker UBP. Similarly, the effects of these blockers could be studied in vivo using the approaches in the studies by Akoake & Aston-Jones [20] and Zhu & Zhou [21]. As one other experimental approach, slices can be generated in which the input from one or several Glu afferent systems (or at least their axons) is preserved as shown for a slice that retains functional connectivity from neurons of the nucleus paragigantocellularis to those in the LC [65]. In such slices, combined with microelectrode array recording [66], it could be studied whether electrical stimulation of these afferents evokes oscillatory activity in neonatal rat LC subregions. Also, in future studies whole-cell V_m_ recording could be combined with dye staining via diffusion from the patch electrode to determine afterwards whether a neonatal LC neuron with particular properties is a projection neuron or rather an interneuron [32]. 

In summary, our present findings are a first mechanistic step in analyzing the role of (AMPAR/KAR-type) iGluRs in LC responses, e.g., to opioids, and (due to its role as a pivotal hub) in their behavioral effects including dependence and withdrawal.

## Figures and Tables

**Figure 1 brainsci-12-00945-f001:**
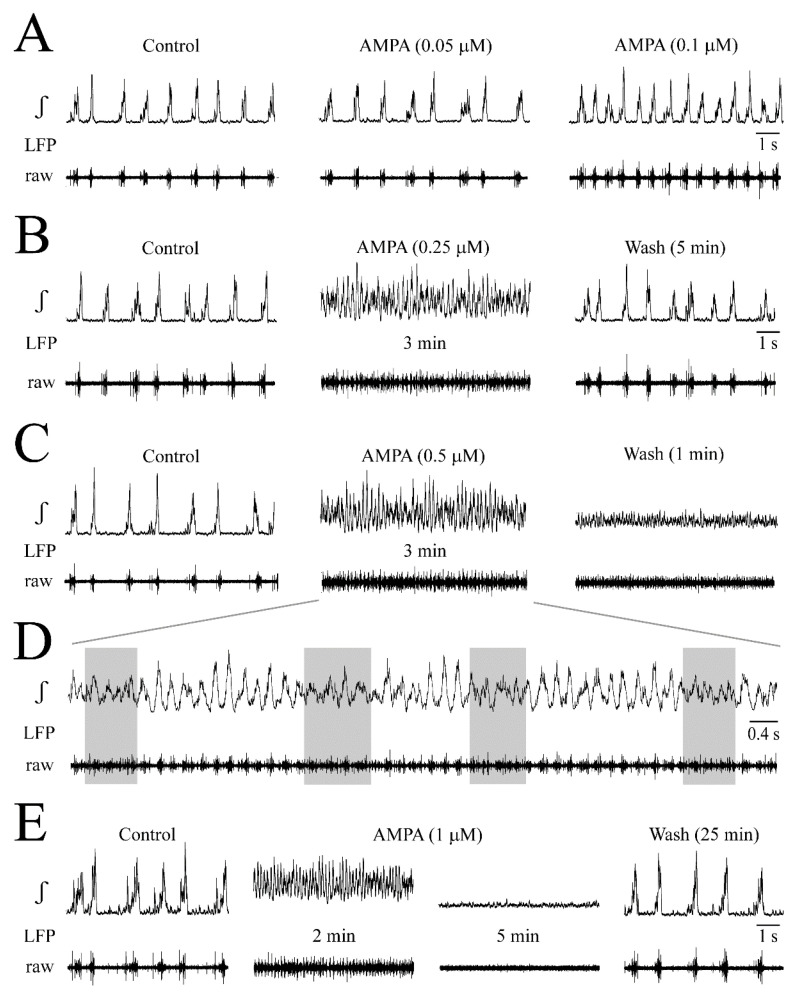
Local field potential (LFP) pattern transformation by the ionotropic glutamate receptor (iGluR) agonist α-amino-3-hydroxy-5-methyl-4-isoxazole propionic acid (AMPA). Bottom traces show the raw extracellular signal and the top trace the integrated (‘moving average’, see integral symbol ∫). Bath-application for 5 min of AMPA slightly accelerated the LFP rate at 0.1 µM (**A**) whereas 0.25 µM merged separate events into faster sinusoidally-shaped oscillations showing every 2–3 s periodic amplitude fluctuations by 10–40% resulting in a spindle-shaped envelope (**B**). (**C**) Within the first 3 min into 0.5 µM AMPA, similar LFP oscillations occurred which then progressively decreased in amplitude and this continued into the early phase of wash. Traces in (**D**) representing signals in (**C**) at higher time resolution show that the smaller amplitude oscillatory events on the integrated trace at 0.5 µM AMPA are due enhanced the tonic discharge that was evident in the raw trace as indicated by the grey boxes. (**E**) After initially similar effects as in 0.25 and 0.5 µM, 1 µM AMPA abolished rhythm which recovered 25 min after the start of the wash. Note that the illustrated wash times vary in this figure and Figures 3, 5 and 7 depending on how quickly the effect of a drug was reversible.

**Figure 2 brainsci-12-00945-f002:**
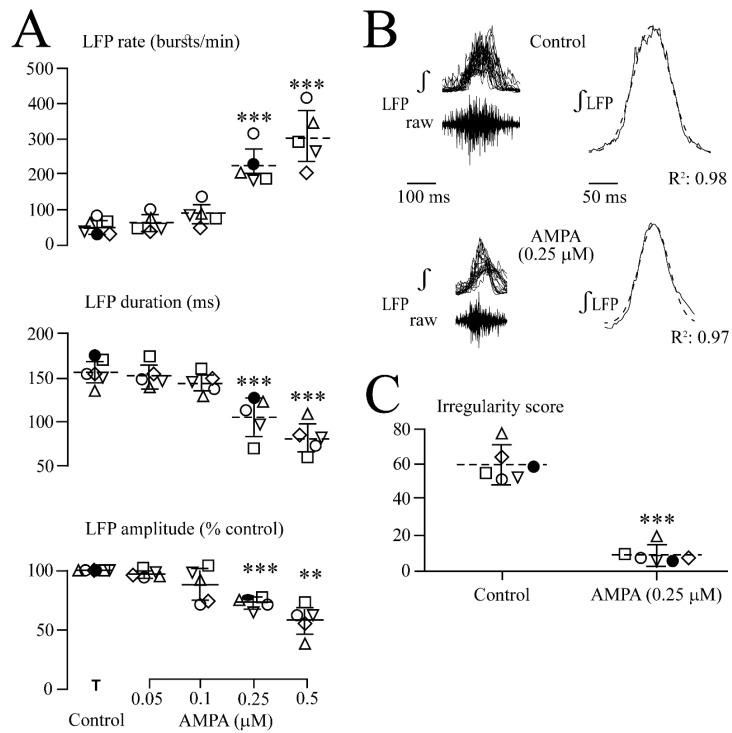
Quantification of AMPA effects on LFP. (**A**) Analysis in five slices (each represented by the same symbol) revealed that the LFP burst rate (upper plots) increased and single event duration (middle plots) and integrated trace amplitude (lower plots) decreased at 0.25 and 0.5 µM. (**B**) Overlay and subsequent averaging of five subsequent bursts from four of the slices in (**A**) revealed a bell-shaped LFP envelope during both the control and 0.25 µM AMPA with a Gaussian fit of R^2^ = 0.98 and R^2^ = 0.97, respectively. (**C**) In six different slices (each represented by the same symbol), 0.25 µM AMPA reduced the irregularity score indicating that rhythm became more regular. The lines indicate mean values (dotted line) ± SD (solid lines); significance (** *p* < 0.01, *** *p* < 0.001) was determined with one-way ANOVA with Dunnett’s post-test in (**A**) (F(1, 5) = 29.64, *p*< 0.0001) (upper plots), (F(1, 5) = 20.42, *p* < 0.0001) (middle plots) and (F(1, 5) = 20.58, *p* < 0.0001) (lower plots) and two-tailed paired t-test in (**C**).

**Figure 3 brainsci-12-00945-f003:**
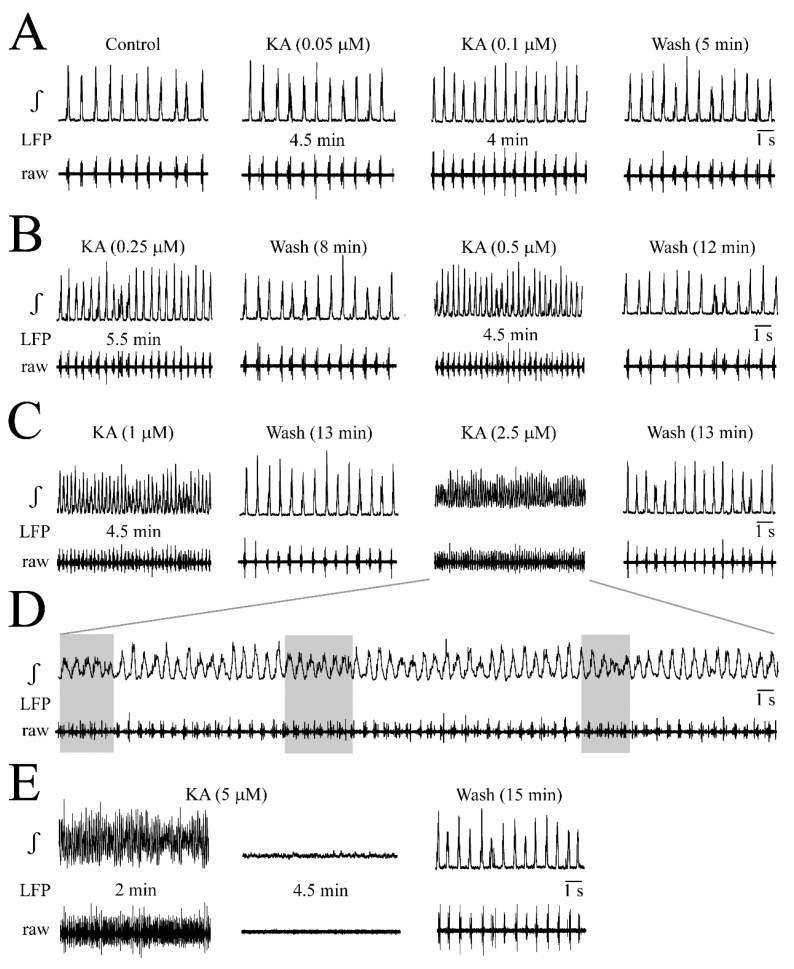
LFP oscillations caused by the iGluR agonist kainate (KA). (**A**,**B**) With no effect of bath-application for 5 min of 0.05 µM KA, both 0.1 and 0.25 µM accelerated LFP events which merged at 0.5 µM into faster sinusoidally-shaped oscillations showing every 2–3 s amplitude fluctuations of 10–40% resulting in a spindle-shaped envelope. (**C**) At 1 µM KA, the rate of spindle-shaped oscillations increased further and these effects were more pronounced at 2.5 µM which also increased the baseline of the integrated signal trace due to tonic spiking. Traces in (**D**) representing signals in (**C**) at higher time resolution show that the smaller amplitude oscillatory events on integrated trace at 2.5 µM KA are due to tonic discharge in the raw trace as indicated by the grey boxes. (**E**) At 5 µM KA, initially spindle-shaped oscillatory events turned within 4.5 min after the start of application into a blockade of rhythm which recovered at 15 min after the start of the wash.

**Figure 4 brainsci-12-00945-f004:**
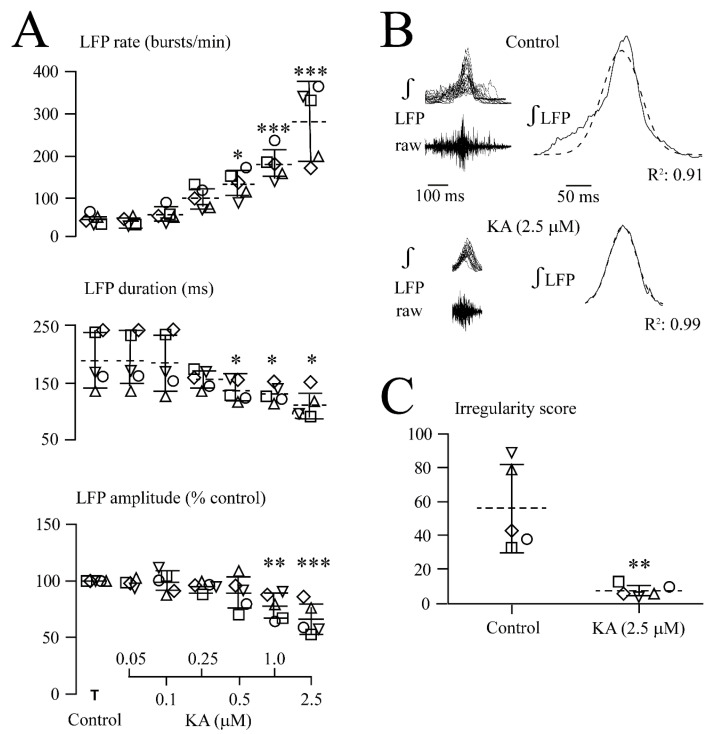
Quantification of KA effects on LFP. (**A**) Analysis in five slices (each represented by the same symbol) revealed that the LFP burst rate (upper plots) increased and single event duration (middle plots) decreased at 0.5, 1.0, and 2.5 µM KA whereas its amplitude decreased only at 1.0 and 2.5 µM (lower plots). (**B**) Overlay and subsequent averaging of five subsequent bursts from four slices revealed a bell-shaped LFP envelope during both control and 2.5 µM KA with a Gaussian fit of R^2^ = 0.91 and R^2^ = 0.99, respectively. (**C**) In five different slices (each represented by the same symbol), the decrease of the IR score shows that 2.5 µM KA made LFP rhythm more regular. The lines indicate mean values (dotted line) ± SD (solid line), significance (* *p* < 0.05, ** *p* < 0.01, *** *p* < 0.001) was determined with a one-way ANOVA with Dunnett’s post-test in (**A**) (F(1, 7) = 20.98, *p* < 0.0001) (upper plots), (F(1, 7) = 4.55, *p* < 0.05) (middle plots) and (F(1, 7) = 7.90, *p* < 0.01) (lower plots) and two-tailed paired *t*-test in (**C**).

**Figure 5 brainsci-12-00945-f005:**
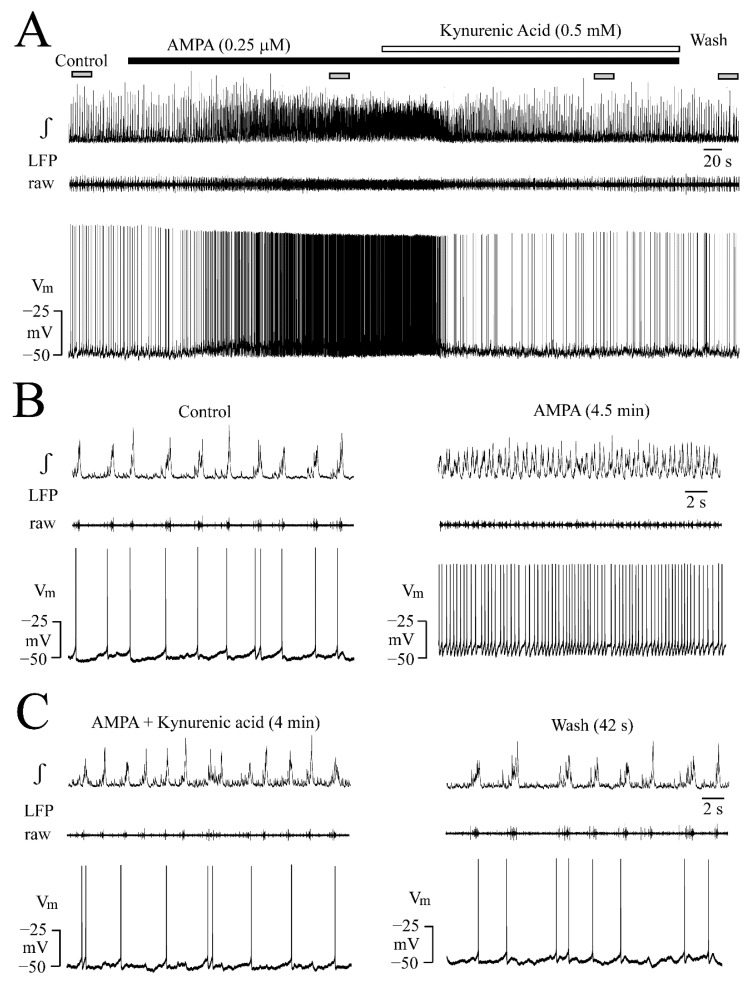
Effects of AMPA and the non-selective iGluR blocker kynurenic acid on LFP and neuronal membrane potential (V_m_). (**A**) The continuous traces show that AMPA accelerated both the LFP (with concomitant pattern transformation) and neuronal spike discharge in association with a depolarization of ~5 mV and that kynurenic acid reversed these effects in the presence of AMPA. (**B**,**C**) The traces on an expanded time scale that were taken at the time periods indicated by the grey boxes in (**A**) show the effects of AMPA and their reversal by kynurenic acid in more detail.

**Figure 6 brainsci-12-00945-f006:**
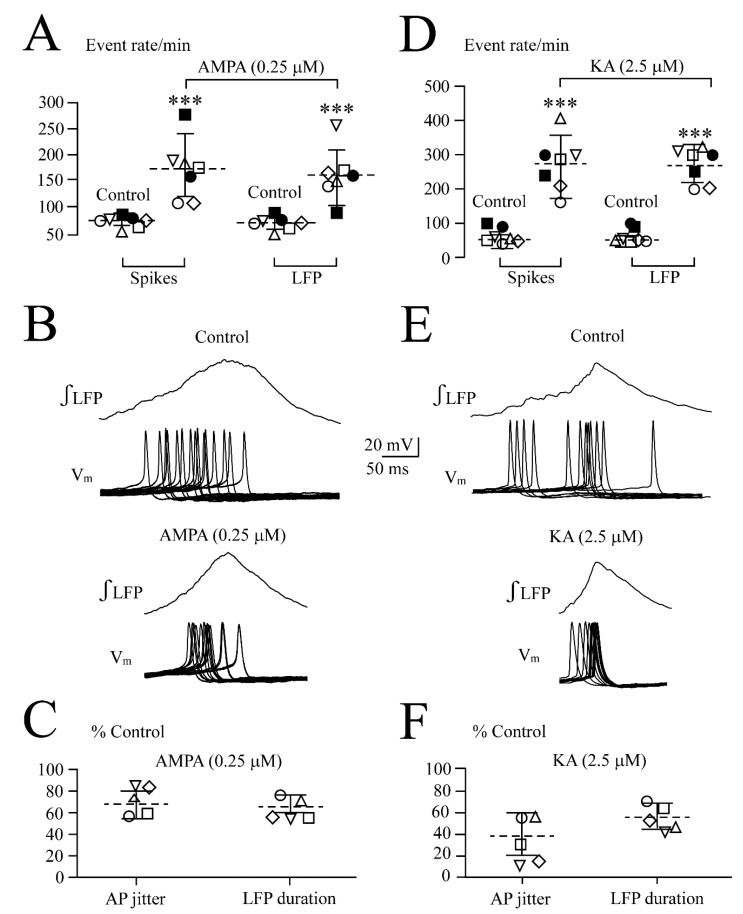
AMPA- and KA-evoked accelerated and less jittered neuronal spiking. (**A**) In seven neurons from different slices (each represented by the same symbol), AMPA similarly accelerated neuronal spiking and LFP rate. The lines indicate the mean values (dotted line) ± SD (solid line). Significance (*** *p* < 0.001) was determined with a one-way ANOVA with Dunnett’s post-test (*** *p* < 0.001). As exemplified for one neuron in (**B**), AMPA reduced spike jitter in a similar fashion as it shortened the LFP event as analyzed in five slices (**C**). (**D**–**F**) show corresponding effects of 2.5 µM KA on different slices (each represented by the same symbol). The lines indicate the mean values (dotted line) ± SD (solid line), significance was determined with two-tailed paired *t*-test.

**Figure 7 brainsci-12-00945-f007:**
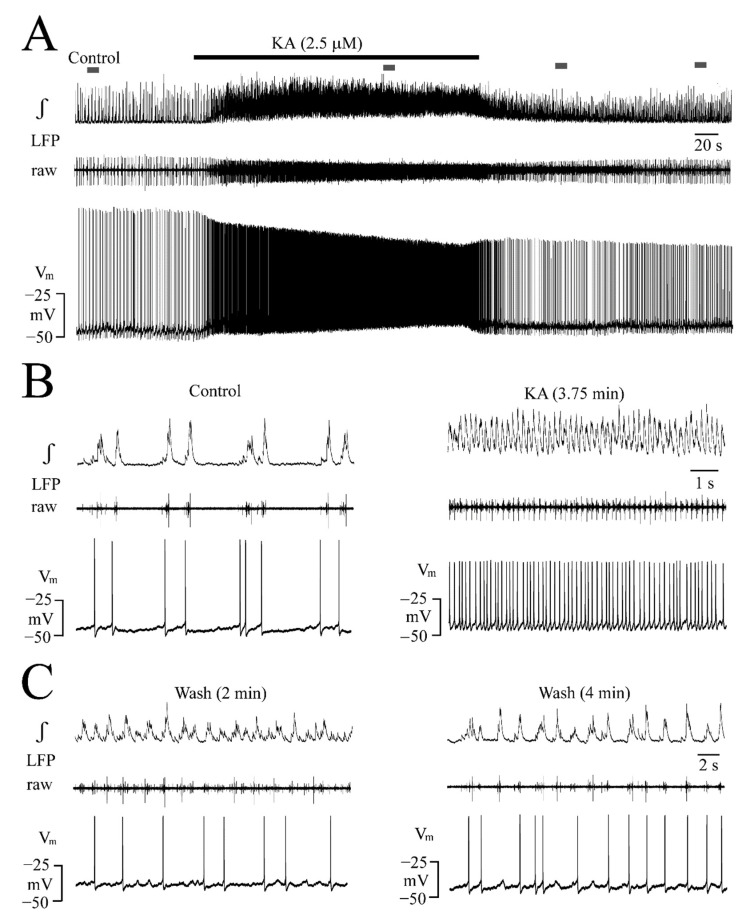
Effects of KA on LFP and neuronal V_m_. (**A**) The continuous traces show that KA accelerated both the LFP (with concomitant pattern transformation) and neuronal spiking in association with a depolarization of ~5 mV. (**B**,**C**) The traces on an expanded time scale that were taken at the time periods indicated by the grey boxes in (**A**) show the KA effects in more detail.

**Figure 8 brainsci-12-00945-f008:**
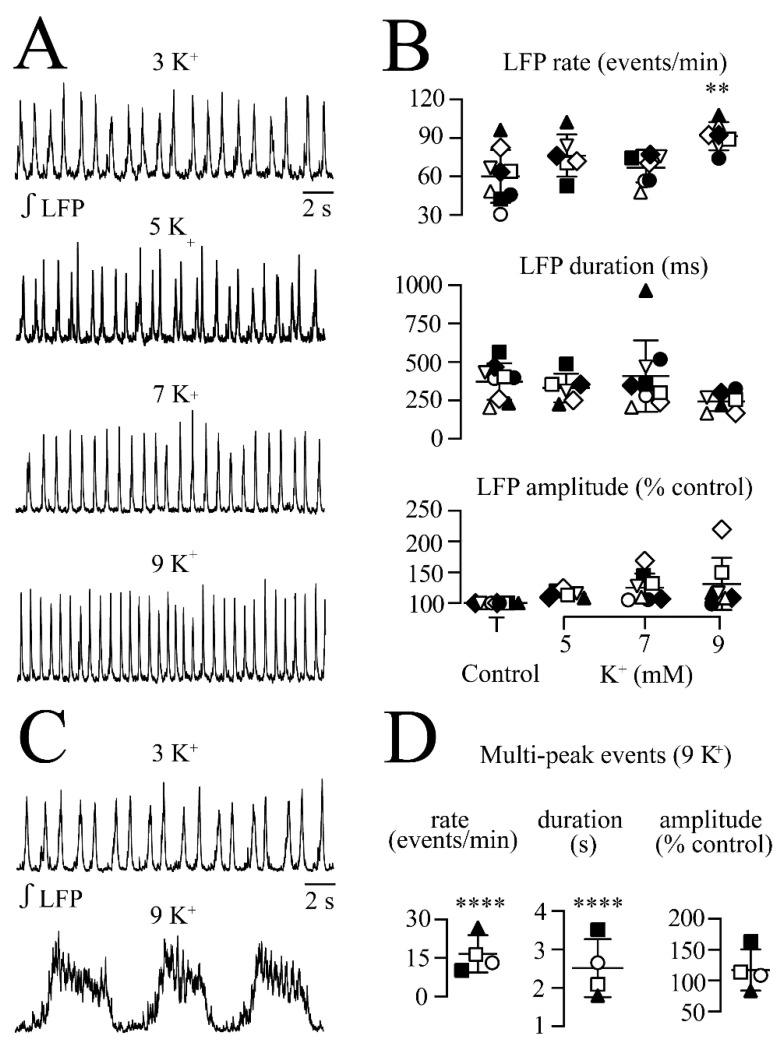
Effects of raised extracellular K^+^ on LFP**.** (**A**) The recordings from a single slice show that raising superfusate K^+^ from physiological 3 mM to 5, 7, or 9 mM shows a trend for accelerating the of the LFP and shortening its duration. (**B**) shows that only 9 mM had a significant effect though, particularly on increasing LFP rate modestly from 60.0 ± 20.7 events/min to 91.4 ± 10.9 events/min, i.e., 1.4-fold. (**C**) exemplifies that in 4 of the 11 slices (each represented by the same symbol), 9 mM K^+^ transformed the LFP pattern. (**D**) Specifically, 9 mM K^+^ slowed the LFP rate 60.0 ± 20.7 events/min to 16.6 ± 7.3 events/min or from 1.0 ± 0.34 Hz to 0.28 ± 0.12 Hz) and transformed the LFP pattern to multipeak events with an increase in its duration from 372 ± 119 ms to 2514 ± 755 ms and no effect on its amplitude. Significance in (**D**) was obtained with two-tailed paired *t*-test. Significances in (**B**,**D**) were ** *p* < 0.01, **** *p* < 0.0001.

**Figure 9 brainsci-12-00945-f009:**
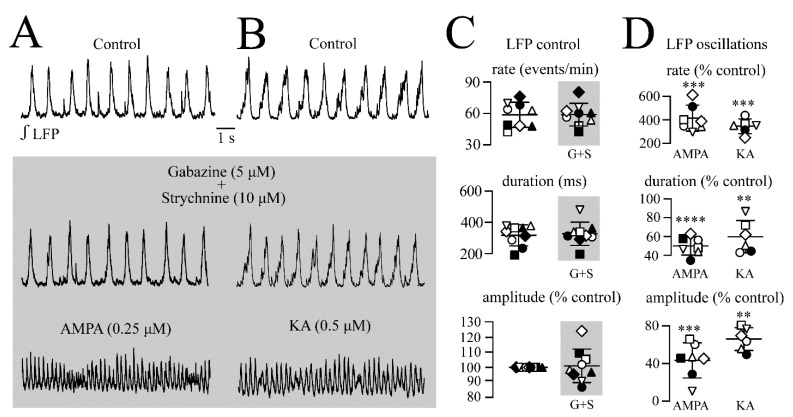
Effects of blockade of Cl^-^-mediated inhibition on the control rhythm and LFP oscillations. (**A**,**B**) The addition of gabazine plus strychnine to the control superfusate had no effect on the LFP rhythms in two different slices and further addition of AMPA (in (**A**)) or KA (in (**B**)) evoked the typical spindle-shaped oscillations. (**C**) summarizes for nine slices (each represented by the same symbol) that gabazine plus strychnine had no effect on the LFP rate, event duration, or the event amplitude. (**D**) Compared to the control rhythm, before adding gabazine plus strychnine, both AMPA (left plots) and KA (right plots) had typical significant effects on the LFP rate, duration, and amplitude. Significance in (**D**) was obtained with two-tailed paired t-test. Significances in (**C**,**D**) were ** *p* < 0.01, *** *p* < 0.001, **** *p* < 0.0001.

**Figure 10 brainsci-12-00945-f010:**
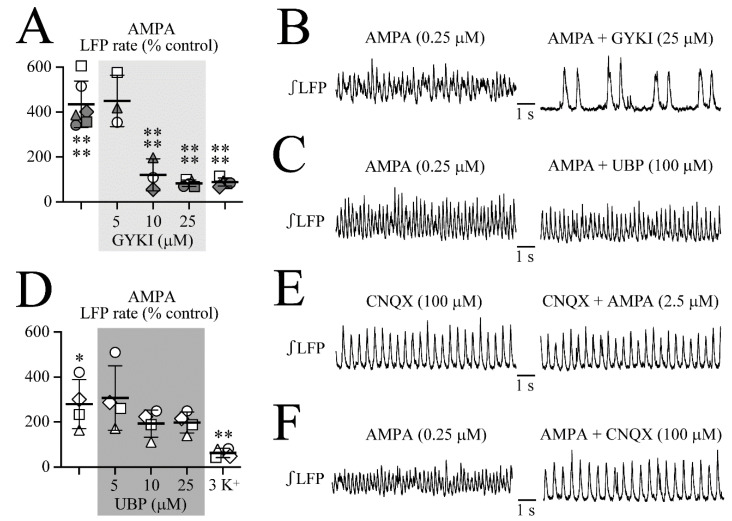
Effects of iGluR blockers on AMPA-evoked LFP oscillations. (**A**,**B**) The LFP oscillations due to either 0.25 or 0.5 µM AMPA were 435% faster than the control rhythm (Each slice is represented by the same symbol). The addition of 5 µM of the selective AMPAR blocker GYKI-53655 did not counter the AMPA-evoked oscillations in contrast to both 10 µM and 25 µM. (**C**,**D**) The AMPA-induced LFP oscillations were not countered by 5, 10, or 25 µM of the selective KAR blocker UBP-302. (**E**) CNQX modestly accelerated the LFP rhythm compared to the control (not shown) but suppressed the spindle-shaped faster oscillations that were typically evoked by AMPA. (**F**) The ‘classical’ AMPAR blocker 6-cyano-7-nitroquinoxaline-2,3-dione (CNQX) acting as partial AMPAR agonist on newborn rat LC neurons [27] blocked the AMPA-evoked LFP oscillations. Significances (* *p* < 0.05, ** *p* < 0.01, **** *p* < 0.0001) in (**A**,**D**) was obtained with one-way ANOVA with Dunnett’s post-test.

**Figure 11 brainsci-12-00945-f011:**
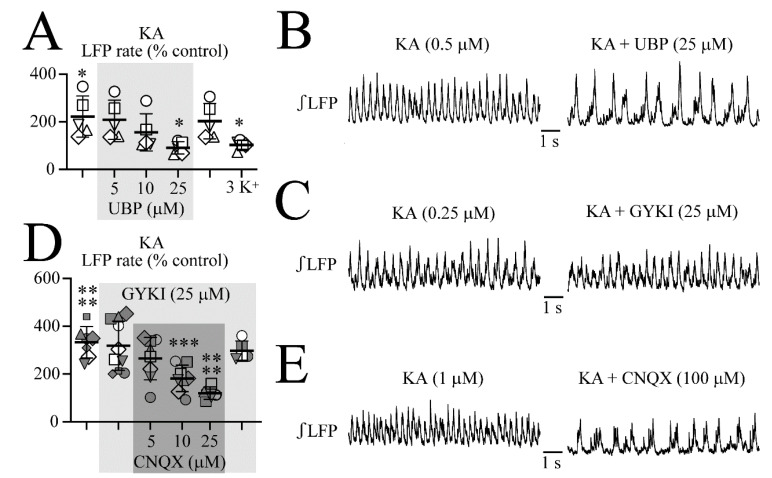
Effects of iGluR blockers on KA-evoked LFP oscillations. (**A**,**B**) The LFP oscillations due to either 0.25 or 0.5 µM KA were 222% faster than the control rhythm. (Each slice is represented by the same symbol.) An addition of 25 µM UBP reversed the LFP oscillations whereas 5 and 10 µM had no effect. UBP depression of KA-evoked LFP oscillations reversed within <10 min after the start of the wash of the blocker and the control rhythm was restored also within <10 min after the start of the KA wash. (**C**,**D**) The KA-induced LFP oscillations were not countered by 25 µM GYKI, but by 10 and 25 µM CNQX whereas 5 µM had no effect. (**D**) The KA-evoked LFP oscillations were also blocked by CNQX in the absence of GYKI. Significances (* *p* < 0.05, *** *p* < 0.001, **** *p* < 0.0001) in (**A**,**D**) were obtained with a one-way ANOVA with Dunnett’s post-test. (**E**) exemplifies that also KA-evoked LFP oscillations in control superfusate without GYKI were countered by CNQX.

## Data Availability

Not applicable.

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
