# Peer review of "Mediation of Sinusoidal Network Oscillations in the Locus Coeruleus of Newborn Rat Slices by Pharmacologically Distinct AMPA and KA Receptors"

_brainsci, 2022, doi:10.3390/brainsci12070945_

Round 1

Reviewer 1 Report

In this manuscript, Rawal and Ballanyi interrogates the cellular mechanisms responsible for the synchrony of locus coeruleus (LC) neurons. To do so, they used acute LC slices of juvenile rats where they found rhythmic activity in the local field potential (LFP), an indirect measure of population activity. They found that activating AMPA/KA receptors created an increase in LFP events and an increase coupling between LFP and LC neuron spikes. They found no effect of blocking Cl- GABA on this AMPA/KA-mediated LFP increase. This represents an interesting step toward understanding the synaptic mechanisms controlling LC population activity.

Major issues:

The abstract and introduction needs to be rewritten. There is not much background provided to put their study in the context of what we know of LC activity in vivo. In the intro, they indicate that LC controls a “variety of brain functions” and that this is “achieved by a modular LC”, but I had a hard time understanding why measuring LFP in slices could give us any additional insight on that. It seems that their main findings: more AMPA/KA activity = more LFP wave-like patterns and more phase-locked LC firing does not really say anything, about asynchronous LC activity which would support modularity. It suggests that activity in the LC is rhythmical and synchronous and becomes even more so with glutamatergic activity. The intro and abstract is very focused on the lab previous results, but I would have appreciated more background on what others have found when recording LFP in vivo or in slice in the LC or surrounding regions. Moreover, abstract first 2-3 sentences should quickly say 1- Background 2- Gap in the knowledge. In the current form, it dives too quickly into methods and specific results from their lab previous work without asking any question or trying to fill a general knowledge gap on LC physiology.

Additionally, intro/discussion need to discuss potential sources for glutamatergic activity in LC. Anatomical studies using CTB (Aston-Jones et al 1991 Prof in Brain Research, Luppi et al 1995 Neuroscience) or targeted rabies virus (Breton-Provencher and Sur 2019 Nat Neurosci, Schwarz et al 2015 Nature) have shown several input regions to LC. Which one could potentially activate AMPA/KA receptors and increase LC LFP rhythmical activity? Which one won’t?

Is LFP they found in newbord rat slices like LFP recorded in vivo?

For the results of Fig 9D, they conclude that blocking GABA activity while applying AMPA/KA does not affect the increase in LFP. Yet, they should compare with the results in Fig 2A and 4A for similar AMPA/KA concentration to support this conclusion. Also for same figure (9D), it is not clear how significance was calculated (compared to control? comparison AMPA versus KA? please specify).

I have several issues regarding statistics, along with my preceding comment on Fig 9D. In methods, they mention that they only performed ANOVA for NMDA effects on LFP, although NMDA was not used in this study. This needs clarification. Also, they do not mention why for similar experiments using the same protocol, N varies in different analysis. This happens throughout the manuscript: Fig 2A N = 5 slices but 2C 6 slices. Fig 6 A,D on 7 neurons but 5 neurons on C,F. Fig 10 and 11 A,D N varies in each condition. Please provide exclusion criteria to justify these discrepancies.  Finally, please specify how significance was obtained in Figs 8D, 9D,10A, 10D, 11A, and 11D.

They mentioned in the intro that LC is just a few thousand neurons, how are they sure they are recording in the LC then? More details on how they targeted and validated their LC recording location is required.

Minor issues:

Do they use a bandpass filter to process their LFP signals? If so, indicate filtering frequencies and filter type in methods.

In methods (Line 181), they say that irregularity score in an established parameter to analyze rhythmic neural response. Please provide citations.

The paper says many times “In another contribution to this special issue”. To my knowledge, the current paper being reviewed has not been accepted yet to the said Special Issue.

Line 210-212: “The amplitude fluctuations during AMPA-evoked LFP oscillations were due to occurrence of tonic spike discharge that was evident in the raw trace and could raise the baseline of the integrated trace if prominent.” Please indicate with arrows this tonic spike discharge directly on Fig 1CD.

They need to indicate how they calculate LFP integral in methods (e.g. do they take a moving average or instant). Also I recommend defining the integral symbol in the figure legend the first time it appears, for clarity.

Please justify why wash times are always different across experiments (Fig 1,3,5,7).

In Fig 8 caption, the description for panel D is missing.

I have found a few typos but please double-check the document for others:

            Line 10: In the LC newborn rat  in the LC of newborn rat

            Line 37: LC in vivo as is also  as it is also the…

            Line 200: Here, firstly dose-response relationships were ….  please rephrase

Reviewer 2 Report

Manuscript brainsci-1772114

Rawal and Ballanyi present an original study on mechanisms of modulation of the firing of neurons located in the region of the locus coeruleus. The manuscript explains in detail the methodology used, highlighting the use of compounds acting on AMPA-type glutamatergic receptors (CNQX, partial agonist, and kainate, KA). The results showed that these AMPA receptor agonists appeared to increase locus coeruleus (LC) network synchrony as indicated by shortening the duration of single local field potential (LFP) events and a corresponding reduction of single-neuron spike jitter. The authors highlight that the pharmacology in the present study described for the first time distinct effects of all these non-NMDA-type iGluR antagonists on a spontaneously active neural network. The results are promising and may bring a new perspective to the intrinsic mechanisms of the neuron circuits on the locus coeruleus.

Specific comments

1. The authors investigated the mechanisms involved in the modulation of neurons located in the LC region, involving activation of non-NMDA glutamate receptors. It would be interesting for the authors to add the origin of afferents to the LC to bring readers information on how the LC neural network is established. In physiological situations, is the tone of LC neurons modulated through interneurons, intrinsic LC neuromodulators, or afferents?

2. Considering the mechanism of the tonic firing of LC neurons, the authors describe the possibility of modulation in the rhythmic release of noradrenaline in the discussion section. However, it is not clear how this modulation occurs. Is there an increase in noradrenaline release or a reduction in this neurotransmission? And for which neural circuits? I emphasize that this point is not the objective of the work, but it could strengthen the discussion of this topic (Consequences for modeling neonatal LC network properties and function).

3. Is it possible to determine whether these LC neurons that can be modulated are projection neurons or interneurons?
